# Therapeutic Perspectives of Molecules from *Urtica dioica* Extracts for Cancer Treatment

**DOI:** 10.3390/molecules24152753

**Published:** 2019-07-29

**Authors:** Sabrina Esposito, Alessandro Bianco, Rosita Russo, Antimo Di Maro, Carla Isernia, Paolo Vincenzo Pedone

**Affiliations:** Department of Environmental, Biological, and Pharmaceutical Sciences and Technologies, University of Campania “Luigi Vanvitelli”, 81100 Caserta, Italy

**Keywords:** *Urtica dioica*, natural products bioactivity, food bioactives, nutraceuticals, cancer therapy, breast cancer

## Abstract

A large range of chronic and degenerative diseases can be prevented through the use of food products and food bioactives. This study reports the health benefits and biological activities of the *Urtica dioica* (*U. dioica*) edible plant, with particular focus on its cancer chemopreventive potential. Numerous studies have attempted to investigate the most efficient anti-cancer therapy with few side effects and high toxicity on cancer cells to overcome the chemoresistance of cancer cells and the adverse effects of current therapies. In this regard, natural products from edible plants have been assessed as sources of anti-cancer agents. In this article, we review current knowledge from studies that have examined the cytotoxic, anti-tumor and anti-metastatic effects of *U. dioica* plant on several human cancers. Special attention has been dedicated to the treatment of breast cancer, the most prevalent cancer among women and one of the main causes of death worldwide. The anti-proliferative and apoptotic effects of *U. dioica* have been demonstrated on different human cancers, investigating the properties of *U. dioica* at cellular and molecular levels. The potent cytotoxicity and anti-cancer activity of the *U. dioica* extracts are due to its bioactive natural products content, including polyphenols which reportedly possess anti-oxidant, anti-mutagenic and anti-proliferative properties. The efficacy of this edible plant to prevent or mitigate human cancers has been demonstrated in laboratory conditions as well as in experimental animal models, paving the way to the development of nutraceuticals for new anti-cancer therapies.

## 1. Introduction

A wide range of chronic and degenerative diseases can be prevented using food product nutraceuticals or functional foods and food bioactive molecules [1,2].

In this context, researchers have studied the *Urtica dioica* [3,4], an evergreen edible plant commonly used since ancient times in traditional medicine to treat several diseases.

*U. dioica* is the most common species of the Urticaceae family commonly known as Stinging nettle and one of the most studied medicinal plants worldwide. It is an herbaceous perennial plant and has a long history of usage for various kinds of health problems [3,4]. The plant grows in tropical and temperate wasteland areas around the world, and well tolerates all environments. The name *Urtica* comes from the Latin verb urere, namely ‘to burn,’ attributed to its stinging hairs. The most common species *dioica* is so defined because the plant generally contains either female or male flowers [5]. The leaves are oval, long petiolate, elongated with toothed margins, the flowers are dioecious, the fruit is a small oval and greenish-yellow achene. The plant has stinging hairs with a tuft of hair at the apex. The leaves and stems contain abundant non-stinging hairs, with touch sensitive tips, needles that will inject chemicals including serotonin, histamine, acetylcholine, moroidin, leukotrienes and possibly formic acid into the skin. The irritant compounds provoke pain, wheals or a stinging sensation [6]. However, this edible plant will lose its irritant powers during cooking, the burning property of the juice is, indeed, dissipated by heat and the young shoots may be used for culinary purposes.

The medicinal properties of *U. dioica* are linked to its anti-inflammatory, anti-asthmatic, astringent, depurative, galactogogue, diuretic, nutritive and stimulating effects. The powered leaf’s extract has been used as an anti-haemorrhagic agent to reduce excessive menstrual flow and nose bleedings. The roots and herbs are used in different ways: the roots for benign prostate hyperplasia, the herbs for urinary tract disorders and rheumatic conditions, while fresh freeze-dried leaves are used to treat allergies [3,4]. Several studies have also reported its analgesic potential and its role as anti-aggregating factor, as well as describing its favorable effects on cardiovascular and smooth-muscle activity as a hypotensive agent [4]. Indeed, the uses of the plant are extended to different fields including the dye industry, veterinary medicine, the textile industry, cosmeceutics for hair loss lotions and anti-dandruff products and also for culinary usage in the preparation of common dishes [7]. In the popular tradition *U. dioica* leaves are eaten, both raw and blanched, gently fried or steamed in many foodstuffs such as pesto, quiches, soups, purées, sauces, cookies, gelatines and jams. Dried herb is processed for capsules, tablets and teas, and other preparations. Freeze-dried herbs are commonly prepared in capsules. Formulations from fresh plant material include homeopathic products, juice and liquid extracts. Targeted studies favored the reuse of *U. dioica* to produce curd for fresh cheese [8]. The proteins (3.7%), dietary fibers (6.4%) and low total calories contents (45.7 kcal/100 g) offer *U. dioica* shoots as a nutritional valuable source and a valid contribution in vitamins (A and C), calcium, iron, sodium and fatty acids [9]. Essential fatty acids, such as linoleic and α-linolenic acid, account for 20.2% and 12.4% of total fatty acids, respectively [10]. Although, the polyunsaturated fatty acids (PUFAs)/saturated fatty acids ratio was reported to be comparably contained, among PUFAs, fatty acids of n-3 and n-6 series were more abundant than monounsaturated fatty acids. This finding was in contrast with a recent investigation regarding fatty acids in *U. dioica* leaves which analyzed a rapid solid–liquid extraction (Soxtherm) using petroleum ether as the solvent, revealing a favorable outcome for saturated acids compared to unsaturated acids [11]. Among monounsaturated fatty acids, a small amount of C17:1 fatty acid was detected (0.13%). Recently, oxylipins, which are bioactive lipid metabolites derived from PUFAs via cycloxygenase (COX), lipoxygenase and cytochrome P450 pathways, were isolated from a *U. dioica* hydroalcoholic extract [12].

Furthermore, *U. dioica* powder was found to be rich in proteins, three-fold higher than traditional cereals, such as rice, wheat and barley [9,13]. *U. dioica* is also rich in minerals, as is characterized by high levels of calcium (169 mg/100 g) and iron (277 mg/100 g) followed by potassium, phosphorus, magnesium, sodium and zinc. According to reported data, *U. dioica* powder is most likely one of the richest source of minerals among the plant foods. The content of carbohydrate is low (37.4%) compared to cereals, such as wheat and barley, showing that *U. dioica* powder has a low glycemic index in relation to the conventional sources of plant foods such as cereals and the potato [13].

The benefits of *U. dioica* may be linked to its diversity in secondary metabolites and its content appears to be strongly influenced by the geographic conditions and taxonomical, morphological and genetics factors [14]. In particular, a number of studies have explored and confirmed *U. dioica* as a favorable source of flavonoids and phenylpropanoids [4].

*U. dioica* may be considered as an infesting plant albeit possessing a longstanding history of medicinal usage. Some medicinal capacities of *U. dioica* have been confirmed by modern research and scientific evidences.

The purpose of this review is to focus, among the many biological activities, on the anti-cancer effects of *U. dioica* by examining the cellular and molecular results of *U. dioica* extract treatments on various human cancer cell lines and in-vivo animal models with special attention to breast cancer, the most prevalent cancer among women.

The study of the mechanisms of action of *U. dioica* may be of importance in the development of food bioactives with natural products for anti-cancer therapies.

PubMed, Scopus and Science Direct were used to consult literature.

## 2. Phytochemical Investigations and Biological Activities of *Urtica dioica*

The phytochemical composition investigation on *U. dioica* [13,14,15,16,17,18,19,20,21,22,23,24,25,26,27,28] revealed that it contains phenolic compounds (including flavonoids, tannins, coumarins and lignans), sterols, fatty acids, polysaccharides and isolectins.

The growing interest in plant phenolic compounds is illustrated by the extensive body of literature devoted to this field of study. Nowadays, it is generally accepted that the therapeutic effects of many plant species are due to the presence of antioxidative phenolics in their tissues. These compounds represent a plant defense mechanism against UV radiation, insects and microorganism, but may also act as plant pigments [29,30,31].

Phenols and polyphenols in dietary plants have gained considerable attention as therapeutic and prophylactic agents in the treatment of chronic and degenerative diseases [32,33]. In particular, it was observed that all the parts (roots, stalk and leaves) of *U. dioica* are a rich source of these substances and that their content is higher in wild plants than in domesticated plants [18,34]. Root samples from Mediterranean cultivar were reported to contain phenol compounds, such ferulic acid and polyphenols as naringin, ellagic acid, myricetin and rutin (Figure 1). The roots also contained lignans (secoisolariciresinol, 9,90-bisacetyl-neo-olivil and their glucosides), phytosterols (e.g., β-sitosterol), polysaccharides, isolectins (mainly *U. dioica* agglutinine), coumarins (e.g., scopoletin), simple phenols (e.g., p-hydroxy-benzaldehyde), triterpenoic acids and monoterpendiols.

*U. dioica* leaves are also constituted by flavonoid glycosides, mainly rutinosyl flavonols, as well as by different depsides of hydroxycinnamic acids with quinic or malic acid (Figure 2). Chlorogenic acid and caffeoyl malic acid represented approximately 76.5% of total phenolic compounds, whereas rutin was the most abundant flavonol derivative [18,19]. Isorhamnetin-3-*O*-rutinoside was found, together with rutin, quercetin-3-*O*-glucoside and kaempferol-3-*O*-glucoside in methanolic extracts of *U. dioica* leaves and stalks [35]. The polyphenol profile seems to be strongly dependent on the parts of the plant investigated, but also on the harvest site and season. The quantification of *U. dioica* phenolics in different extracts, by high-performance liquid chromatography coupled with tandem mass spectrometric detection, evidenced that inflorescence extracts were the richest extracts [17]. Thus, the consumption of *U. dioica* was in line with an amelioration of phenolic compounds food intake and thus, the exploitation of these anti-oxidant and anti-inflammatory compounds defined the plant as a valuable tool towards mutagenesis and carcinogenesis [36].

Among lipid secondary metabolites, carotenoids were detected in the leaves and their total content was estimated equal to 29.6 mg/100 g dry weight [4,23].

Several studies have established that extracts of *U. dioica* possess various pharmacological effects [37,38,39], including anti-inflammatory [40,41,42], anti-oxidant [43,44,45,46,47], anti-microbial [46,47,48,49], anti-diabetic [39,50,51], cardiovascular [39], anti-ulcer, analgesic [36], immuno-modulatory [35,52], anti-mutagenic [44] and anti-cancer properties. Moreover, this edible plant has considerable chemopreventive capacities and disease-preventing effects on animals and humans. These health benefits of *U. dioica* may be related to the wide range of bioactive natural products present in the various parts of the plant.

The preventive activity of both polyphenols and carotenoids is associated to the health promoting effects of *U. dioica* against chronic and degenerative diseases such as cancer.

## 3. Anticancer Activities of *Urtica dioica*

Among the biological activities of *U. dioica*, we report in detail the studies on the anti-cancer effects, due to the induction or inhibition of the key processes in cellular metabolism and the ability to activate the apoptotic pathways. Various studies have recently demonstrated the cytotoxic and anti-cancer properties of *U. dioica*, in particular against colon, gastric, lung, prostate and breast cancers. In this section, we will review the main anti-tumour activities of *U. dioica* demonstrated against several human cancer cell lines and in animal models.

Cancer is a group of diseases in which normal cells grow uncontrollably and abnormally, invade and spread to other parts of the body. Unfortunately, it is a main cause of death worldwide and the incidence and mortality rates are still unacceptably high [53,54,55,56]. Notable progresses have been obtained in conventional therapies (as chemotherapy, radiotherapy and surgical excision), but these treatments cause many serious side effects and often may prolong life for only a few years.

Cancer chemoprevention [57,58] has become an important therapeutic option through which the battle against cancer could be possible, using natural, synthetic or biological agents able to reverse, suppress or prevent either carcinogenesis or the progression of premalignant cells towards invasive tumors. For this purpose, plants and herbs may be a promising source for adjuvant, complementary or alternative anti-cancer therapy [59,60,61,62,63,64,65,66,67,68], since some of them contain bioactive natural products and anti-cancer compounds, including polyphenols [32,33,69,70] as flavonoids, tannin etc. Today, many phytochemical compounds, normally biosynthesized and accumulated in the plants, have shown chemopreventive actions and several anti-cancer drugs, including podophilotoxins, camptotoxins, taxans, arise from herbal compounds and are successfully used for anti-cancer therapy [63,71]. Chemoprevention due to the natural plant-based bioactivity can be achieved through different biochemical and molecular mechanisms involved in cancer control and development. The plants, indeed, contain a number of bioactive molecules that are able to induce cellular protection and responses to stresses, such as anti-oxidant enzymes, apoptosis and/or cell cycle arrest [72,73].

The anti-mutagenic activity of a protein fraction from the aerial parts of *U. dioica* was demonstrated, via the Ames test in various bacteria strains, against the mutagen 2-aminoanthracene; the anti-mutagenic activity can be due to the inhibition of CYP450-isoenzymes, involved in the 2-aminoanthracene mutagen bioactivation [44].

The anti-oxidant and radical scavenging activities of *U. dioica* were reported by ABTS and superoxide-radical scavenger assays and on analysis of the changes in antioxidant enzymes [43,44,45,46,47]. In mice, the treatment with *U. dioica* methanolic extract from the aerial parts demonstrated hepatoprotective and nephroprotective activities against cisplatin-induced [74,75] toxicity, most likely due to increasing antioxidant defense mechanisms, in fact, this extract has been shown to increase the activities of catalase (CAT) and superoxide dismutase (SOD) enzymes and the content of glutathione (GSH) [76]. These activities may be attributed to the flavonoid content of *U. dioica*, in fact, the flavonoids were associated to anti-oxidant and radical scavenging activities [77,78,79].

The reactive oxygen metabolites have a well-known role in cancer pathogenesis [80,81]. The oxidative stress, with the loss of cellular redox homeostasis and elevated levels of oxygen free radicals, causes the production of mutagenic agents and can be tumorigenic, with a key role in initiation and progression of cancer [82]. Due to the presence of large quantities of compounds with anti-oxidant and free radical scavenger properties, *U. dioica*, and specifically the leaves, are able to reduce the high level of oxidative stress present in cancerous cells and exert a chemopreventive function.

Most drugs in use for cancer treatment are cytotoxic and/or cytostatic [83]. Since the rate of apoptosis was reduced during cancer in several studies, an efficient cancer treatment requires the induction of apoptosis in cancer cells, with the programmed cell death of cancerous and damaged cells, without destructive adverse effects in normal dividing cells [84]. Furthermore one of the more significant properties analyzed in putative cytotoxic anti-tumor agents is the ability to induce apoptosis and/or cell cycle arrest. The cytotoxic activity of *U. dioica* was widely tested by in vitro MTT assay and trypan blue viability exclusion dye assay (to evaluate the number of live cells). The cellular and molecular mechanisms of toxicity were analyzed using different assays. The most commonly used were DNA fragmentation assay and TUNEL test (to detect the type of cell death, apoptosis or necrosis), quantitative Real-Time PCR (qRT-PCR, to quantify the apoptosis- and metastasis- related mRNA expression levels), Western Blotting (to quantify apoptosis-related protein levels) and flow cytometry (to analyze cell cycle distribution and apoptosis). The Table 1, Table 2 and Table 4 (Table 4 is included in the next paragraph) summarize the main findings regarding the anti-cancer properties of *U. dioica* with the plant extracts and parts used (specifying collection sites and the biologically active molecules identified), the cancer cell lines/tissues or animal models tested, the IC50 (concentration required for 50% inhibition) and the effects. The studies (first author and date) are chronologically listed for various tumoral groups. The studies on benign prostatic hyperplasia are also reported.

In the study by Ghasemi et al. (2016) [86] the cytotoxic effects of an ethanolic extract of *U. dioica* roots (0–2000 μg/mL) were demonstrated on human colon (HT29) and gastric (MKN45) cancer cells. Cells were treated with increasing concentrations of *U. dioica* for 24–72 h. *U. dioica* decreased cell viability in a dose- and time- dependent manner, with IC50 values of 24.7 and 249.9 μg/mL, respectively, after 72 h exposure. In addition, *U. dioica* treatment induced apoptotic cell death, as shown by flow cytometry analysis. The different studied cell lines showed a diverse sensitivity to the *U. dioica* treatment, with more sensitive human colon cancer HT29 cells compared to human gastric cancer MKN45 cells, a cell line poorly differentiated and usually resistant to chemotherapy. Interestingly, the anti-proliferative effects of *U. dioica* treatments are comparable to those obtained with oxaliplatin, a current anti-neoplastic drug. In line with these findings, Mohammadi et al. (2016) [87], demonstrated the cytotoxic effects of a dichloromethane extract of *U. dioica* aerial parts plant on human colon cancer cell line HCT-116, with IC50 of 23.61 μg/mL (48 h treatment) and by eliciting apoptotic cell death and arresting the cell cycle at the G2/M phase.

The dichloromethane extract of *U. dioica* leaves inhibited the growth and proliferation of human prostate cancer cells (PC3), showing a IC50 concentration of 15.54 μg/mL in 48 h exposure and a cell cycle arrest in G2/M phase (Mohammadi et al. 2016) [92]. In this study, the observed increased expression levels of pro-apoptotic genes caspase 3 and 9 and reduced anti-apoptotic Bcl-2 suggested that cytotoxicity was due to apoptosis induction from intrinsic (mitochondrial) pathway. Moreover, a methanolic extract of *U. dioica* roots previously produced a significant dose- and time- dependent reduction in proliferation of human prostate carcinoma cells (LNCaP), with a 30% maximum growth inhibition after 5 day exposure with 1 μg/mL extract concentration (Konrad et al. 2000) [89].

More recently, D’Abrosca et al. (2019) [12] reported the effects of *U. dioica* leaves methanolic extract against the human non-small cell lung cancer cell lines (NSCLC). Exposure of H1299 and A549 NSCLC cells to this extract inhibited cell proliferation, with an IC50 of 52.3 and 47.5 μg/mL, respectively. NSCLC cells have a low sensitivity to cisplatin [74,75,93], a cytotoxic agent largely utilized for chemotherapy cancer cure. The co-treatment with the *U. dioica* extract and cisplatin ameliorated the cisplatin cytotoxicity, thus showing a synergistic effect. *U. dioica* extract induced arrest at G2/M cell cycle phase and apoptosis from extrinsic pathway, as demonstrated by the observed decreased levels of pro-caspase 3 and pro-caspase 8 proteins (indicating the activation and increasing of the apoptotic proteolytic enzymes caspase 3 and caspase 8) and increased levels of cPARP and tBid (substrates of caspase 3 and caspase 8, respectively). GADD153 (a marker of endoplasmic reticulum stress) [94,95,96] and DR5 (death receptor [97] a cell surface receptor of the TNF-receptor superfamily, which directly promotes the extrinsic apoptotic pathway) were also upregulated after *U. dioica* treatment, confirming that the extracts promoted the extrinsic apoptotic pathway. Interestingly, rutin and oxylipins (polyunsaturated oxidised fatty acids) [98,99,100] were identified in the *U. dioica* extract on investigation of the exact mechanism involved in cell death, via spectroscopic techniques NMR [101,102] and mass spectrometry analyses. In particular, oxylipins, including the most abundant 13-S-hydroxy-9Z, 11E, 15Z-octadecantrienoic acid, were proved to be the bioactive natural products responsible for anti-cancer activity (Figure 3 and Table 3). Oxylipins are a large and diverse family of secondary metabolites derived from the oxidation of PUFAs [103]. The oxylipin 13-S-hydroxy-9Z, 11E, 15Z-octadecantrienoic acid also possess anti-inflammatory properties in human chondrocytes [41]. It is noteworthy that a plant oxylipin, 12-oxo-phytodienoic acid, potently suppressed the proliferation of human breast cancer cell lines T47-D and MDA-MB-231, reducing the expression of the cyclin D1 and inducing arrest at G1 phase of the cell cycle [104,105]. In addition, another biologically active molecule of *U. dioica* is a rare lectin (carbohydrate-binding protein), *U. dioica* agglutinine (UDA, accession number P11218 in Protein Data Bank) [106], isolated from the *U. dioica* aqueous roots extract (Wagner et al. 1994) [85]. This molecule demonstrated in vitro anti-proliferative properties on human cervical cancer HeLa cells and human epidermoid carcinoma A431 cancer cells. On HeLa and A431 cells UDA inhibited the binding of EGF to its receptor with an IC50 of 5 and 21 μg/mL, respectively, after 24 h exposure [107].

Moreover, in-vitro studies and investigations regarding testosterone-induced rat models of prostatic hyperplasia, examined the effects of petroleum ether and ethanolic extracts of the *U. dioica* roots (28 day, 50 mg/Kg, oral treatment), Nahata et al. (2012) [91]. The results demonstrated that *U. dioica* significantly reduced the activity of 5α-reductase enzyme and the dimension of the prostatic hyperplasia. 5α-reductase [108] is a key enzyme involved in testosterone metabolism, thus in hormone-dependent prostate hyperplasia and prostate cancer. The IC50 values for 5α-reductase were of 0.19 and 0.12 mg/mL, respectively, on the 28^th^ day of treatment. In both the extracts, via spectroscopic techniques, the β-sitosterol was isolated; in ethanolic extract scopoletin was also found (Figure 3 and Table 3). β-sitosterol is a sterol originated by a complex and multistage biosynthetic process [109]; it was jet reported to treat patients with prostate diseases [110,111]. Scopoletin is a courmarin derived by a known biosynthesis pathway [112]; it possesses anti-inflammatory properties [113] and has been reported to induce anti-proliferative and pro-apoptotic effects on the prostate cancer cell line PC3 [114]. In a previous animal study (Lichius et al. 1999) [88], the experimentally induced benign prostatic hyperplasia in Balb/c mouse was reduced, by a methanolic extract of *U. dioica* roots, orally administered for 28 days (5 mg), with an observed 51.4% growth inhibition.

Figure 3 present the molecular structures of specific biologically active compounds identified from *U dioica*. Table 3 summarizes the biological activities of specific bioactive molecules isolated from the *U dioica* extracts in each anti-cancer study.

### Urtica dioica and Breast Cancer

Breast cancer is the most prevalent cancer among women and one of the main causes of death worldwide. Breast cancer statistics indicate that in the US one woman in eight will suffer from breast cancer and that more than 200,000 new patients with breast cancer will be diagnosed every year [115,116]. Tumor invasion and metastasis remain the main causes of patients’ mortality and still present an important therapeutic challenge. For the treatment of breast cancer, a multidisciplinary approach is currently used, involving surgery, radiotherapy, chemotherapy, hormone therapy, immunotherapy and other novel treatment strategies such as gene silencing, but they are associated with serious side effects [56,117,118]. Various new therapeutic targets for adjuvant, complementary and alternative medicines, including natural products bioactivity from plants, have been proposed by several new studies to treat breast cancer patients [60,62,119].

In this paragraph, we report investigations, via in-vitro studies and animal models, on *U. dioica* as a potential natural source of food bioactives and chemotherapeutic agent for breast cancer (Table 4).

In a study by Fattahi et al. (2013) [45], the activity of *U. dioica* leaves aqueous extract was analyzed on the human breast cancer cell line MCF-7. The *U. dioica* extract demonstrated anti-oxidant and anti-proliferative activity. After 24, 48 and 72 h of exposure to different concentrations of the *U. dioica* extract, significant cell death was observed in a dose-dependent manner, with an IC50 value of 2 mg/mL concentration after 72 h of treatment. In accordance with previous observations regarding anticancer drugs [118], the decrease of cell viability caused by *U. dioica* was due to the induction of apoptosis but not to necrosis. The treatment-induced apoptosis was demonstrated at the cellular level by morphological observation, DNA ladder formation, flow cytometry analysis and at the molecular level by measuring the increased amount of the different apoptotic-related proteins caspase 3, caspase 9, Bax (a pro-apoptotic protein), Bcl-2, calpain 1 (a calcium-dependent cytosolic cysteine protease) and calpastatin (a specific inhibitor of calpain 1). Noteworthy was the increase in the anti-apoptotic Bcl-2 protein; indeed, Bcl-2 can interact with Nur 77/TR3 and convert to Bax-like death effector, subsequently inducing apoptosis [120]. These findings were in agreement with a recent study by Fattahi et al. (2018) [121]. In this study, two human breast cancer cell lines, MCF-7 (estrogen and progesterone receptors positive; wild-type P53) and MDA-MB-231 (estrogen and progesterone receptors negative; mutated P53), were treated with *U. dioica* aqueous extract of leaves. The cytotoxicity of the treatment was confirmed, with IC50 values for both breast cancer cell lines of approximately 2 mg/mL, after 72 h exposure. *U. dioica* extract induced the apoptosis and increased the expression levels of Bax, especially in MCF-7 cells. Interestingly, *U. dioica* extract has been shown to influence the gene expression of two other proteins, adenosine deaminase (ADA) [122] and ornithine decarboxylase (ODC1) [123]. The expression level of ADA gene in MCF-7 cell line was increased in a dose-dependent manner, but did not modify in the MDA-MB-231 cell line. Alternatively, the ODC1 gene was upregulated in both cell lines. These enzymes show a regulatory role in cellular processes such as proliferation, cell growth and apoptosis [123,124]. In particular, ADA is a key enzyme in adenosine metabolism and nucleotide DNA turnover; ODC1 is the key enzyme in biosynthesis of polyamines [125,126]. Considering that *U. dioica* extracts contain phytoestrogens [127], the differences observed in these two cell lines could be due to a diverse status of hormone receptors. Moreover, it is possible that *U. dioica* in MDA-MB-231 cells may induce apoptosis via a P53-independent pathway.

Furthermore, it is interesting to report that the *U. dioica* leaves aqueous extract in the prostate tissue of patients with prostate cancer, while, inhibited the activity of ADA, with IC50 of about 50 μg/mL (calculated, 30 min treatment) (Durak et al. 2004) [90].

Different modes of actions and effects of the *U. dioica* extracts are then possible in the various cell lines and tissues. Differences in in-vitro conditions related to patient’s tissue and cell lines also to be considered.

Mohammadi et al. (2016) [129] in a previous study also demonstrated the cytotoxic and apoptotic effects of a dichloromethane extract of *U. dioica* leaves, in MDA-MB-468 cells, a human breast adenocarcinoma cell line. The dichloromethane extract of *U. dioica* induced a dose- and time-dependent anti-proliferative effect. The IC50 concentrations were of 29.46 and 15.54 μg/mL for 24 and 48 h exposure, respectively. These experimental values for IC50 demonstrate that the dichloromethane extract of *U. dioica* leaves have a more potent cytotoxic effect, on MDA-MB-468 cells. In this cell line, *U. dioica* dichloromethane extract caused cell death through apoptosis as revealed by morphological changes, TUNEL test, DNA fragmentation ladders and mRNA expression levels of apoptotic-related genes. In particular, *U. dioica* activated apoptosis through the intrinsic pathway, as revealed by the increase in the caspase 3 and caspase 9, the decrease in the Bcl2 and any significant changes in caspase 8 expression levels. Interestingly, Mohammadi et al. (2016) [130] demonstrated a synergic effect on cell death and invasion of human breast cancer MDA-MB-468 cell line, by treatment of *U. dioica* leaves dichloromethane extract in combination with the paclitaxel drug. Paclitaxel is one of the most commonly used natural drugs (derived from the bark of pacific yew tree) approved for chemotherapy in different types of cancers, such as ovary cancer, breast cancer and non-small cell lung cancer, acting as an anti-microtubule chemotherapy drug [134]. The antitumor potency of combinational therapy with paclitaxel and *U. dioica* extract was investigated on the human breast cancer cell line MDA-MB-468, demonstrating that *U. dioica* significantly increased the sensitivity of breast cancer cells to paclitaxel therapy, ameliorating its cytotoxicity. In fact, the MTT test demonstrated, in a time- and dose-dependent manner, a strong reduction of cell viability and of IC50 values for paclitaxel in the co-treatment with *U. dioica* extract: from 6.73 μM for paclitaxel alone, to 0.59 μM for co-treatment, after 24 h. The synergic effect of *U. dioica* extract and paclitaxel was also demonstrated on cell migration; in fact, by scratch test [135], a decreased invasion rate and a reduced number of migrated cells were observed. The molecular mechanism involved was elucidated studying the synergic effect of *U. dioica* and paclitaxel on the expression of snail-1 and related genes ZEB1, ZEB2 and twist. Snail-1 is a protein involved in invasion and migration of cancer cells [136,137,138] and is required for metastatic ability in breast cancer [139]; in fact, the silencing of snail-1 gene using specific siRNA prevented the metastasis of breast cancer cells. The observed reduction of the expression of snail-1 and related genes after the co-treatment was in agreement with its anti-metastatic potential. Moreover, a synergic effect of *U. dioica* and paclitaxel on the cell cycle arrest also revealed cell cycle arrest occurring at the G2/M phase, with a decreased Cdc2 expression. In agreement with these studies, Mansoori et al. (2017) [131], demonstrated that the dichloromethane extract of *U. dioica* leaves significantly decreased the cell proliferation of three different breast cancer cell lines, the human MCF-7 and MDA-MB-231 and mouse 4T1. The observed IC50 concentration of *U. dioica* extract was 31.37 mg/mL in MCF-7, 38.14 mg/mL in MDA-MB-231 and 35.21 mg/mL in 4T1 cells, at 48 h of treatment. Moreover, the scratch assay demonstrated an inhibitory effect of *U. dioica* on the migration of the breast cancer cell lines. Moreover, the authors investigated the signalling pathway by which *U. dioica* could inhibit the cell migration. In detail, demonstrated that *U. dioica* extract could inhibit tumor metastasis by regulating miR-21 (a crucial oncomir that is overexpressed in advanced tumors and metastasis) [140,141,142,143,144,145,146], the matrix metalloproteinases [147] MMP1, MMP9, MMP13, the vimentin [148], CXCR4 [149,150] and E-cadherin [151], important metastasis-related genes involved in cellular invasion by modifying adhesion junctions and the migratory capacity of cells [152]. In particular, miR-21, MMP1, MMP9, MMP13, CXCR4 and vimentin were found overexpressed in the invasive margins of breast cancer tissues of clinical samples and in the cancer cell lines; E-Cadherin, on the other hand, was decreased. *U. dioica* extract treatment decreased miR-21 expression, which substantially reduced the overexpressed MMP1, MMP9, MMP13, vimentin and CXCR4, and increased E-cadherin in the treated tumor cell lines. In a previous study (Abu-Dahab and Afifi, 2007) [128] the cytotoxic effects of *U. dioica* ethanol extracts from leaves and stems were tested on human breast cancer cell lines MCF-7 and the percentage of MCF-7 cell survival after 72 hrs exposure to 50 μg/mL extract was 93.12%.

Finally, the *U. dioica* root methanolic extract inhibited, via an in-vitro test, aromatase enzyme activity in a concentration dependent manner [153]. Aromatase is a key enzyme involved in steroid hormone metabolism (mediating the conversion of androgens into estrogens) and is targeted in hormonal therapy of hormone-sensible breast cancers, thus acting on the cancer promotion.

Taken together, the various studies on cell lines demonstrated a diverse sensitivity to the *U. dioica* treatments and the extract tested showed differences in anti-cancer potency. It is important to consider that the cell survival and IC50 discrepancies observed in the various studies could be caused by differences in the habitat and parts of the plant used, in cell line types investigated and in the *U. dioica* extraction process.

The effectiveness of *U. dioica* to treat breast cancer has been proved not only in laboratory conditions but also in in-vivo experimental models.

Mohammadi et al. (2017) [132] prepared an *in vivo*-induced model of breast cancer, mice Balb/c with allograft tumors caused by injecting subcutaneously 4T1 murine breast tumor cells, and then administered *U. dioica* leaves dichloromethane extract (10 or 20 mg/kg body weight) by intraperitoneal injection for 28 days. Interestingly, *U. dioica* extract significantly reduced the tumor masses in the treated mice and significantly diminished the size and weight of the tumors removed from the treated mice. By TUNEL assay, it was shown that the *U. dioica* extract induced apoptosis in Balb/c allograft tumor models. Furthermore, the Ki-67 test demonstrated that the *U. dioica* treatment reduced tumor growth, decreasing the percentage of cell proliferation in the breast cancer tissue. Then, real-time PCR studies revealed that the intraperitoneal injection of *U. dioica* extract into the model mice was able to elicit the intrinsic pathway of apoptosis with increased expression of pro-apoptotic caspase 3 and caspase 9 and a downregulation of anti-apoptotic Bcl2. In the treated mice (4T1-induced Balb/c mouse model of breast cancer), according to the results previously reported on the breast cancer cell lines MCF-7, MDA-MB-231 and 4T1, *U. dioica* treatment induced the anti-metastatic pathway, with decreased expression of miR-21, MMP1, MMP9, MMP13, vimentin and CXCR4 and increased expression of E-cadherin (Mansoori et al. 2017) [131].

Finally, in an animal study by Telo et al. (2017) [133], the effects of *U. dioica* in N-methyl-N-nitrosourea-induced rat model of breast cancer were investigated. Aqueous extract of *U. dioica,* 50 g/kg powdered, was added into the food of rats for 5.5 months. The lipid peroxidation, the antioxidant enzyme activities and the formation of mammary gland cancer was then evaluated. *U. dioica* administration decreased the levels of lipid peroxidation and increased catalase antioxidant enzyme activity in rats generated mammary tumors. The results demonstrated, besides, a reduced rate in formation of breast cancer, with a decreased number of cancer masses.

In Figure 4 we summarize the main anti-cancer studies and results reported for the various *U. dioica* extracts and cancer types.

Each *U. dioica* extract and its effects are identified using different color codes. The dashed arrows denote the results obtained with whole extracts; the solid arrows indicate the specific molecules isolated from the extracts and their effects. A generic picture of apoptosis signalling, indicating where and how molecules or extracts of *U. dioica* act on the different cellular targets, is included. By different images are specified if these results were observed in-vitro or in-vivo, on specific cancer types.

## 4. Conclusions and Future Perspectives

Cancer has become the second most frequent cause of death after cardiac diseases and recent analyses provide an increase of its prevalence in the near future. Breast cancer is the second cause of cancer death among women. Currently, a variety of treatments such as chemotherapy, radiotherapy, hormone therapy and surgery, as well as newer nanotechnology and gene silencing therapy, are used in the treatment of cancer, but induce side effects. Hence, the need to develop the most effective anti-cancer therapy with few side effects and high cytotoxicity that will effectively arrest the initiation and progression of the cancer.

In recent years, many researchers have analyzed natural products and low cost drugs for cancer cure and prevent cancer development. Plants are a precious source of anti-cancer agents; the use of plants for cancer treatment is popular in many Asian cultures and today, beneficial compounds from these plants are used in the production of different modern anti-cancer drugs. Recent studies have illustrated that adjuvant therapy with natural products could help to prevent the development of cancer, as well as cure and improve the survival rate of patients.

Several studies have shown the anti-cancer properties of *U. dioica*, however, to the best of our knowledge, no previous report has reviewed the effects of *U. dioica* extracts on different cancer cell lines and animal cancer models.

Taken together, the main studies on the anti-cancer ability of *U. dioica* extracts provide a promising chance for the use of *U. dioica* as a nutraceutical food for the prevention and treatment of several cancers, including breast cancer. The various extracts of *U. dioica* tested, in fact, prevent cancerogenesis, kill human cancer cells and inhibit their migration. The extracts were not toxic and differences in the growth of the cancer cells was observed compared to the controls (untreated cancer cells) and normal cells, indicating their safety and a promising strategy to reduce adverse effects and ameliorate the efficacy of cancer chemotherapies.

*U. dioica* may exert biological anti-cancer activities through various mechanisms of actions, including antioxidant and anti-mutagenic properties, induction or inhibition of key processes in cellular metabolism and ability to activate the apoptotic pathways. Most anti-cancer drugs induce apoptosis, as a primary mechanism for inhibition of cell proliferation. The apoptotic effect of *U. dioica* in cancer cell lines and animal models was studied at the cellular and molecular levels. The type of cell death (apoptosis or necrosis) was investigated and the pathways involved in apoptosis induction were demonstrated, by studying the genes and proteins involved in the apoptosis process.

The *U. dioica* extracts contain varied bioactive molecules and the ability of these extracts to treat cancer, is due to those components that inhibit tumor growth and induce the apoptosis pathway.

The principal bioactivity of *U. dioica* was found in lipophilic fractions (e.g., dichloromethane extracts), suggesting that lipophilic compounds are mostly responsible for the anti-cancer actions. From the lipophilic fractions phytosterols, pentacyclic triterpenoids, coumarins, ceramides and hydroxyl fatty acids were isolated. In the hydrophobic extracts of the roots the sterols stigmast-4-en-3-one, stigmasterol and campesterol are present, which inhibited the enzyme activity of Na+, K+-ATPase in patients’ tissues with benign prostatic hyperplasia and may subsequently repress prostate-cell metabolism and proliferation. The hydrophilic fractions (e.g., water, methanol, ethanol extracts) of *U. dioica* also demonstrate a high bioactivity, indicating that even polar active principles are responsible for the anti-cancer activity. The hydrophilic fractions contain isolectins and some polysaccharides. In the polar extracts of the roots the lignans as (+)-neoolivil, (-)-secoisolariciresinol, dehydrodiconiferyl alcohol, isolariciresinol, pinoresinol and 3,4-divanillyltetrahydrofuran are also present.

The most likely explanation for the considerable anti-cancer effect of *U. dioica* is the content of flavonoids and other known molecules and/or still unknown substances. Among the food bioactive molecules of *U. dioica*, the flavonoids are polyphenolic compounds that are able to induce anti-cancer effects through different mechanisms such as anti-oxidant activity, induction of apoptosis, inhibition of cell growth and cell migration. In fact, several plants rich in flavonoids possess disease preventive and therapeutic properties and, in particular, the consumption of vegetable and fruit rich in flavonoids is associated with reduced cancer risk.

Therefore, *U. dioica* may be used as a nutraceutical food bioactive in cancer treatment to prevent or reduce cancer without presenting the side effects of current anti-cancer treatments.

However, the effects observed could be caused by several molecules and, probably, molecules that have not yet been identified. Further studies are required to isolate and characterize the pure bioactive molecules in this plant to better understand its multiple anti-cancer actions and to explore these potentials in the fight against human cancers.

## Figures and Tables

**Figure 1 molecules-24-02753-f001:**
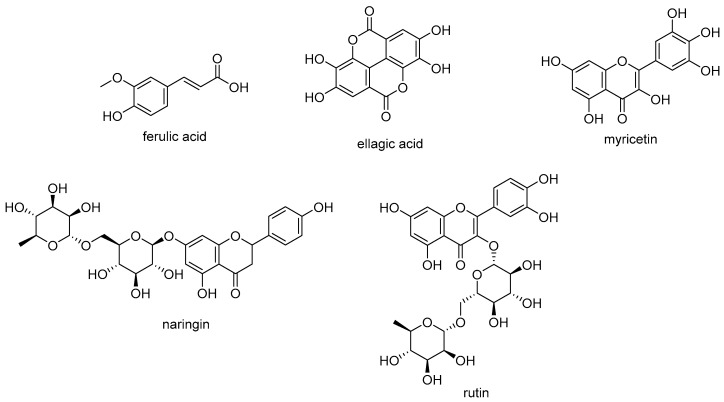
Phenolic compounds from *U. dioica* roots.

**Figure 2 molecules-24-02753-f002:**
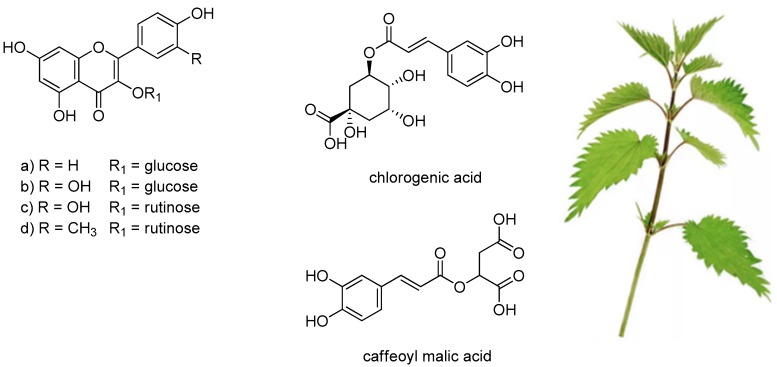
Phenols and polyphenols mostly detected in *U. dioica* leaves. (**a**) kaempferol-3-*O*-glucoside; (**b**) quercetin-3-*O*-glucoside; (**c**) rutin; (**d**) isorhamnetin-3-*O*-rutinoside.

**Figure 3 molecules-24-02753-f003:**
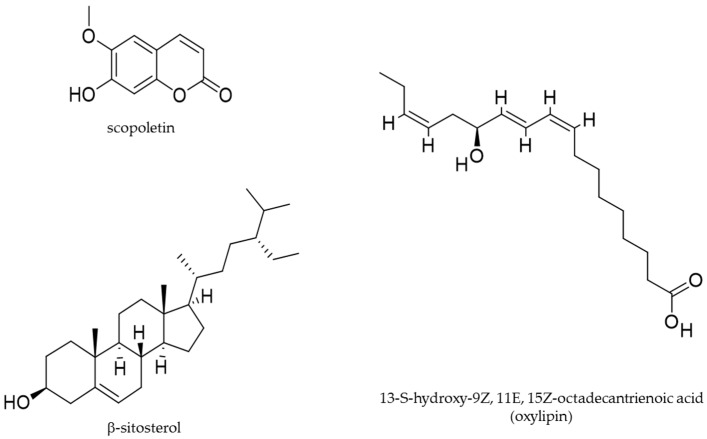
Molecular structures of selected bioactive phytochemicals isolated from *U. dioica*.

**Figure 4 molecules-24-02753-f004:**
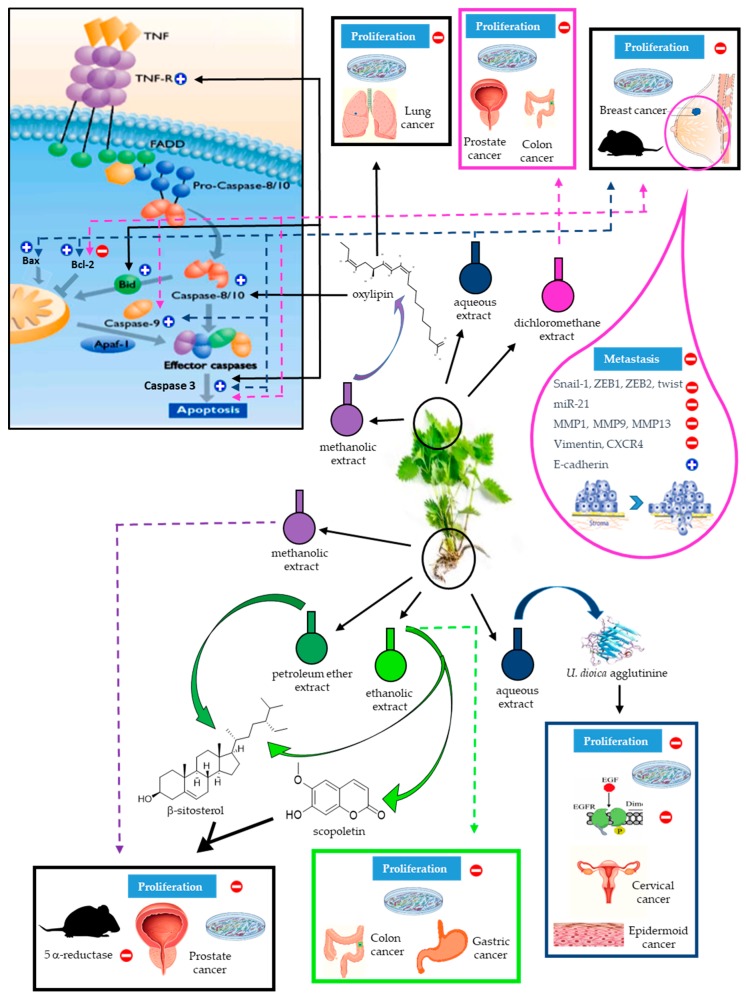
Schematic drawing of *U. dioica* anti-cancer effects. Each *U. dioica* extract and its effects are identified using different color codes. The dashed arrows denote the results obtained with whole extracts; the solid arrows indicate the specific molecules isolated from the extracts and their effects. A generic picture of apoptosis signalling, indicating where and how molecules or extracts of *U. dioica* act on the different cellular targets, is included. By different images are specified if these results were observed in-vitro or in-vivo, on specific cancer types.

**Table 1 molecules-24-02753-t001:** Anti-cancer activities of *U. dioica*: cervical, epidermoid, colon, gastric and lung cancer.

*U. dioica* Extracts *	Cancer Cell Lines	IC50	Effects	References
*U. dioica L.* (Germany) roots aqueous extract *U. dioica* agglutinine (UDA)	HeLa human cervical cancer A431 human epidermoid carcinoma	5 μg/mL (24 h treatment) 21 μg/mL (24 h treatment)	↓ Proliferation ↓ EGF binding	Wagner 1994 [85]
*U. dioica L.* (Iran) roots ethanolic extract	HT29 human colon cancer MKN45 human gastric cancer	24.7 μg/mL (72 h treatment) 249.9 μg/mL (72 h treatment)	↓ Proliferation ↑ Apoptosis	Ghasemi 2016 [86]
*U. dioica L.* (Iran) aerial parts dichloromethane extract	HCT-116 human colon cancer	23.61 μg/mL (48 h treatment)	↓ Proliferation ↑ Apoptosis G2/M arrest	Mohammadi 2016 [87]
*U. dioica L.* (Italy) leaves methanolic extract, oxylipins	NSCLC H1299 human non-small cell lung cancer NSCLC A549 human non-small cell lung cancer	52.3 μg/mL (72 h treatment) 47.5 μg/mL (72 h treatment)	↓ Proliferation ↑ Apoptosis extrinsic pathway ↑ caspase 3 ↑ caspase 8 ↑ cPARP ↑ tBid ↑ GADD153 ↑ DR5 G2/M arrest	D’Abrosca 2019 [12]

* The plant extracts and parts used with specified collection sites and the biologically active molecules identified.

**Table 2 molecules-24-02753-t002:** Anti-cancer effects of *U. dioica*: prostate cancer, in-vitro and in-vivo studies.

*U. dioica* Extracts *	Cancer Cell Lines/Tissues/Animal Models	IC50	Effects	References
*U. dioica L.*(Germany) roots methanolic extract	Balb/c mouse model of benign prostatic hyperplasia (28 days, 5 mg oral treatment)		↓ hyperplasia 51.4 % growth inhibition	Lichius 1997 [88]
*U. dioica L.*(Germany) roots methanolic extract	LNCaP human prostate cancer		↓ Proliferation 30% (5 day treatment with 1 μg/mL)	Konrad 2000 [89]
*U. dioica L.* leaves aqueous extract	prostate tissue from prostate cancer patients	50 μg/mL (30 min treatment)	↓ ADA	Durak 2004 [90]
*U. dioica L.* (India) roots petroleum ether extract, β-sitosterol	rat model of benign prostatic hyperplasia (28 days, 50 mg/Kg oral treatment)	0.19 mg/mL (28 day treatment)	↓ hyperplasia ↓ 5α-reductase	Nahata 2012 [91]
roots ethanolic extract, β-sitosterol and scopoletin		0.12 mg/mL (28 day treatment)		
*U. dioica L.* (Iran) leaves dichloromethane extract	PC3 human prostate cancer	15.54 μg/mL (48 h treatment)	↓ Proliferation ↑ Apoptosis intrinsic pathway ↑ caspase 3 ↑ caspase 9 ↓ Bcl-2 G2/M arrest	Mohammadi 2016 [92]

* The plant extracts and parts used with specified collection sites and the biologically active molecules identified. ADA: adenosine deaminase.

**Table 3 molecules-24-02753-t003:** Biological activities of specific molecules isolated from *U. dioica*.

Molecules	Biological Activities	Cells	References
13-S-hydroxy-9Z, 11E, 15Z-octadecantrienoic acid (oxylipin)	Anti-proliferation ro-apoptotis Stop cell cycle Anti-inflammation	lung cancer chondrocytes	[12,41]
*U. dioica* agglutinine (UDA)	Anti-proliferation Anti-EGF binding	cervical cancer epidermoid carcinoma	[85,107]
β-sitosterol	Anti-proliferation Inhibition 5α-reductase	prostate	[91,110,111]
scopoletin	Anti-proliferation Inhibition 5α-reductase Anti-inflammation Pro-apoptotis	prostate prostate cancer	[91,113,114]

**Table 4 molecules-24-02753-t004:** Anti-cancer effects of *U. dioica*: breast cancer, in-vitro and in-vivo studies.

*U. dioica* Extracts *	Cancer Cell Lines/ Animal Models	IC50	Effects	References
*U. dioica L.* (Jordan) leaves and stems ethanol extract	MCF-7 human breast cancer		↓ Proliferation 7% (72 h treatment with 50 μg/mL)	Abu-Dahab 2007 [128]
*U. dioica, L.* (Iran) leaves aqueous extract	MCF-7 human breast cancer	2 mg/mL (72 h treatment)	↓ Proliferation ↑ Apoptosis intrinsic pathway ↑ caspase 3 ↑ caspase 9 ↑ Bax ↑ Bcl-2 ↑ calpain 1 ↑ calpastatin	Fattahi 2013 [45]
*U. dioica, L.* (Iran) leaves dichloromethane extract	MDA-MB-468 human breast cancer	15.54 μg/mL (48 h treatment)	↓ Proliferation ↑ Apoptosis intrinsic pathway ↑ caspase 3 ↑ caspase 9 ↓ Bcl-2	Mohammadi 2016 [129]
*U. dioica, L.* (Iran) leaves dichloromethane extract	MDA-MB-468 human breast cancer	0.59 μM (24 h co-treatment paclitaxel + extract)	↓ Proliferation ↑ Apoptosis ↓ Migration ↓ Snail-1 ↓ ZEB1, ZEB2, twist G2/M arrest ↓ Cdc2	Mohammadi 2016 [130]
*U. dioica, L.* (Iran) leaves dichloromethane extract	MCF-7 human breast cancer MDA-MB-231human breast cancer 4T1 mouse breast cancer Balb/c mouse model of breast cancer (28 day, 20 mg/Kg injection treatment)	31.37 mg/mL (48 h treatment) 38.14 mg/mL (48 h treatment) 35.21 mg/mL (48 h treatment)	↓ Proliferation ↓ Migration ↓ miR-21 ↓ MMP1, MMP9, MMP13, vimentin, CXCR4 ↑ E-cadherin	Mansoori 2017 [131]
*U. dioica, L.* (Iran) leaves dichloromethane extract	Balb/c mouse model of breast cancer (28 day, 20 mg/Kg injection treatment)		↓ Metastasis ↑ Apoptosis intrinsic pathway ↑ caspase 3 ↑ caspase 9 ↓ Bcl-2 ↓ Ki-67	Mohammadi 2017 [132]
*U. dioica,* aqueous extract	rat model of breast cancer (5.5 months, 50 g/kg food treatment)		↓ Metastasis ↓ lipid peroxidation ↑ catalase	Telo 2017 [133]
*U. dioica, L.* (Iran) leaves aqueous extract	MCF-7 human breast cancer	2 mg/mL (72 h treatment)	↓ Proliferation ↑ Apoptosis ↑ ADA ↑ ODC1	Fattahi 2018 [121]
	MDA-MB-231 human breast cancer	2 mg/mL (72 h treatment)	↓ Proliferation ↑ Apoptosis = ADA ↑ ODC1	

* The plant extracts and parts used with specified collection sites. ADA: adenosine deaminase; ODC1: ornithine decarboxylase.

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
