# Peer review of "Therapeutic Perspectives of Molecules from Urtica dioica Extracts for Cancer Treatment"

_molecules, 2019, doi:10.3390/molecules24152753_

Round 1

Reviewer 1 Report

This review submitted by Esposito et al., entitled “Therapeutic perspectives of Urtica dioica extracts for cancer treatment” describes the potential of Stinging nettle plant extracts for anticancer therapy in addition to conventional treatments which have side effects.

The manuscript is clear and well written but some typos and grammar mistakes must be corrected. I consider that at the present time the manuscript requires some corrections, additional information and modifications before acceptance.

Editing:

Line 2: Considering that you also mentioned and discussed molecules in your review, I would recommend to include it too in the title.

Line 44: “has a long history of usage for various kinds of health problems”, please include references.

Line 45: “weathers”, considering that you are talking about growing conditions, I would recommend to use the term “environments” or “climates”.

Line 102: Please correct “vary”.

Line 135: Please check “isoquercetrin”.

Line 419: “The breast cancer”, please delete “The”.

Line 436: Please correct “extract”.

Line 441 to 444: Please change for “U. dioica may exert biological anti-cancer activities by a variety of different mechanisms of actions, among which antioxidant and anti-mutagenic properties, induction or inhibition of key process in cellular metabolism and ability to activate the apoptotic pathways.”

Comments:

This review is well written and give a lot of information, however I regret that only some information are summarized and presented in a large table and there is not a real work of compiling information and providing a review of all information and references that you mentioned in your manuscript.

Figure 1: Please provide names of the different molecules and not only the variable substitutions.

Table 1: Is there a reason why only some data are summarized in the table? For example, line 274 to 280, why these information, results are not presented in the table?

Same comment starting line 285, why do not you compile these information on a table?

I want also recommend to include some figures in your review, summarizing results and information. For example a generic picture of apoptosis signaling and pointing out where and how molecules or extracts of Urtica dioica acts on the different cellular targets. Using different color codes or signs you can also specify if these results were observed in-vitro or in vivo, on specific cellular types,…. It would prove a real work of synthesis for your review.

Line 259, 260, 265: you mentioned several specific compounds or rare lectins present in Urtica dioica, why do not you present their molecular structures too?

Reviewer 2 Report

This is a review summarizing health benefits and biological activities of Urtica dioica extracts with a particular focus on its cytotoxic, anti-tumor and anti-metastatic effects against several cancers in vitro and in vivo. This review paper may be useful to both natural product chemists and biologist. However, the main problem with the manuscript is that it uses bad grammars and has too many typos. Therefore, significant polishing is necessary before publishing. 

Content-wise, the authors are suggested to link the biological activities of Urtica dioica to specific chemicals (if demonstrated in literatures) from the extracts in each case study and summarize them in a table, which will be helpful to natural product chemists. The authors should also summarize or cite literatures that report known biosynthetic/synthetic pathways of the phytochemicals isolated from Urtica dioica, which could be useful to both biochemists and chemists.

Round 2

Reviewer 1 Report

This review submitted by Esposito et al., deals about the potential of extracts and molecules from Urtica dioica and their use in cancer therapy. After a first reviewing, my comments and remarks were taken into account by the authors and I consider that at the present time the manuscript requires some minor modifications and correction before acceptance.

Editing:

Page2: Please check

“The benefits of U. dioica may be linked to its diversity in secondary metabolitesand its content”.

Page 3: Please check

samples from Mediterranean cultivar was reported to contain phenol compounds, such ferulic acid”.

Page 4, Figure 2

Please correct your “OR” variable residue, considering a and b molecules (R = OH). 

Page 16, legend

"A generic picture of apoptosis signalling, indicating where and how molecules or extracts of U. dioica acts on the different cellular targets, is included".